# Pinpointing Crowd in Bird's Eye View via Proximal Contexts

## Abstract

Bird's-Eye-View (BEV) perception has emerged as a promising paradigm for scene understanding and it has recently been applied for multi-human activity analysis. We formulate multi-human BEV standing localization as a regression task under severe inter-occlusion in crowded scenes. To address this, we propose a unified dual-space collaborative learning framework, denoted as BEVCrowd-Locator, that jointly reasons over image and BEV spaces to augment monocular inference. We incorporate standing point queries that explicitly model spatial relationships between adjacent individuals as proximal contexts, disambiguating foot-keypoint localization via proximity-aware attention. Additionally, we present the proximity-aware suppression mechanism that prioritizes consensus predictions during regression, improving robustness to occlusions. We validate our model's performance in terms of social distance measurement on the real-world surveillance benchmark. Moreover, we quantify the BEV localization performance on real soccer videos, demonstrating its potential in sports analysis.

## 1 Introduction

Bird's Eye View (BEV) perception has been widely studied across diverse fields due to its ability to provide a comprehensive, top-down perspective of a social scene. In autonomous driving, BEV perception is essential for tasks such as localization, obstacle detection, and path planning. By integrating data from sensors like cameras or LiDAR, BEV perception generates a detailed view of the surroundings, enabling precise high-definition map reconstruction and navigation (Nishimura et al., 2023a). Recently, BEV has also been applied in surveillance (e.g., for social distance measurement (Dai et al., 2021)) and sports analytics (e.g., for tracking player positions (Somers et al., 2024)).

We address the problem of localizing individual people for multi-human scenes in a bird's-eye view from perspective images. Unlike crowd density estimation, our work targets instance-level localization, which is essential for applications such as surveillance in shopping malls or player tracking in sports. Among the works most relevant to ours, Dai et al. (2021) proposes an image-to-BEV crowd analysis model that transforms a surveillance camera's perspective into an overhead BEV view, enabling social distance measurement within crowded settings. This approach relies on crowd density estimation via a fully convolutional network (CNN) to approximate individual locations. However, the generated density maps are typically too coarse to reliably distinguish closely positioned individuals in dense crowds, which significantly reduces the accuracy of the image-to-BEV transformation. Furthermore, this density-based approach is not well-suited for scenarios where precise localization of each individual is required.

To address these limitations, we introduce a novel multi-human localization model, BEVCrowd-Locator, tailored for dynamic environments where individual-level localization is critical. Unlike crowd-based models that rely on density maps, BEVCrowdLocator directly predicts individual BEV coordinates, bypassing density estimation altogether. In our formulation, given an image of a multi-human activity scene, the task is to predict each person's standing location in the image space, which is mapped to BEV space. This approach is particularly well-suited for sports and similar settings, where fine-grained positioning of individuals is essential.

Regressing standing locations in monocular images faces a dilemma: identifying head positions (easier to detect but height-sensitive) or feet locations (occlusion-prone but BEV-projection-stable).

While heads are visually salient, height estimation errors introduce significant BEV projection deviations. Conversely, feet prediction is inherently ill-posed due to frequent occlusion but benefits from contextual cues: human body priors (single-person) and neighboring interactions (group scenarios). Thus, we propose a context-driven feet localization approach that leverages these cues to infer positions under limited visibility. We also demonstrate in experiments that our approach outperforms head detection under occlusion, validating its robustness.

To implement this approach, we establish a collaborative dual-space learning paradigm. With the primary goal of directly regressing individuals' standing locations in image space, we develop an auxiliary network to incorporate contextual information from BEV space. The supervisory information provided by BEV annotations allows the model to bypass occlusion challenges, adaptively compensating for missed predictions during the image-space learning phase. Additionally, we set the embeddings of individual standing points as trainable queries within both branches of our model, enabling the network to learn contextual features of standing locations from both image and BEV perspectives. These queries facilitate the estimation of approximate foot locations based on proximal contexts (e.g., human body priors and neighboring interactions), particularly in cases of limited visibility. More importantly, by identifying relationships among queries for different standing points, the model can effectively distinguish between the foot locations of neighboring individuals. Despite the benefits of using proximal context, the risk of misidentifying a single individual's position as multiple distinct points may persist. To address this, we enhance standing point regression in the image space by incorporating proximity-aware suppression. This technique aggregates similar points, reducing the influence of redundant estimates that correspond to the same individual, thereby improving localization accuracy.

For evaluation, we demonstrate the superior performance and efficiency of our model in multi-human localization and social distance measurement on the CityUHK-X-BEV benchmark (Dai et al., 2021). Additionally, we incorporate a trivial cross-frame association module that enables BEVCrowdLocator to perform point-based multi-human tracking in BEV space, quantifying its performance on a sports video benchmark, SoccerNet (Cioppa et al., 2022). Despite the simplicity of the association strategy, our approach reconstructs individual trajectories smoothly and robustly without relying on pretrained detector, showcasing its potential for sports analysis. Furthermore, we provide supplementary videos highlighting the BEV multi-human tracking results, featuring smooth and continuous trajectories. The key contributions of this paper are as follows:

- To address the challenge of multi-human BEV localization, we formulate the task as a standing location prediction problem. This challenge, primarily caused by occluded visibility, is mitigated through a collaborative dual-space learning scheme that compensates image-space inference with BEV contextual information.

- We introduce standing point queries as trainable embeddings, enabling it to learn proximal context effectively and differentiate the foot positions among neighboring individuals.

- To mitigate repeated detections of the same individual, we incorporate proximity-aware suppression into standing point regression, improving crowd localization accuracy.

- We validate our superior performance in multi-human localization and social distance measurement on the CityUHK-X-BEV benchmark. Additionally, we showcase the performance of our method on soccer videos without relying on pretrained human detector.

## 2  RELATED WORKS

**BEV Scene Understanding.** The traditional Bird's Eye View (BEV) transformation task (Hou et al., 2020; Hou & Zheng, 2021; Yang et al., 2021; Hu et al., 2021; Zhou & Krähenbühl, 2022; Saha et al., 2022; Zhao et al., 2024; Liu et al., 2024) mainly involves converting the camera's view into a top-down view for scene understanding, widely applied in fields like autonomous driving (Man et al., 2023; Lin et al., 2024; Feng & Sun, 2024; Ding et al., 2024; Jiang et al., 2023) for understanding crowds or vehicles. These methods typically rely on ground plane keypoints, multi-view images, or perspective transformations. However, in complex and dynamic crowd environments, traditional BEV methods struggle with issues such as occlusion and the diverse interaction behaviors of individuals. While some methods attempt to address these challenges by analyzing multiple views, obtaining such multi-view information is often infeasible in crowded scenarios.

Figure 1: Illustration of our proposed BEVCrowdLocator framework. It comprises a primary branch that regresses the multi-human standing locations in the image space via a hybrid CNN-transformer based framework, as well as an auxiliary branch that leverages BEV contexts to compensate the primary task through the BEV-to-image projection. In addition, we set the embeddings of individual standing points as the trainable queries of both branches, in order to learn the proximal contextual features of standing locations. Last, we incorporate proximity-aware suppression into the point matching scheme to achieve multi-human standing locations.

**BEV-based Crowd Analysis.** As the major topics in crowd analysis, crowd counting and localization methods have been extensively investigated in recent years (Liu et al., 2019a; Babu Sam et al., 2017; Zhang et al., 2016; Sam et al., 2020; Liu et al., 2019c; Wang et al., 2021; Liang et al., 2022a; Song et al., 2021a; Xu et al., 2022; Liang et al., 2022b; Gao et al., 2020; Han et al., 2023; Jiang et al., 2019b;a; Song et al., 2021b; Liu et al., 2023; Han et al., 2021; Qian et al., 2024; Han et al., 2022). In recent years, BEV-based crowd analysis has emerged, and its purpose is to project people's locations onto a bird's eye view (Nishimura et al., 2022; 2023b; Dai et al., 2021; Dendorfer et al., 2022). For instance, Dai et al. (2021) uses a multi-branch convolutional network to predict the density distribution of head and feet for a crowd. Comparing to BEV-Net, we formulate this problem as a task of standing location regression, which can provide more precise positioning than coarse density and can be applied to videos. Dendorfer et al. (2022) present a multi-object tracking method leveraging trajectory prediction in BEV. Given a monocular video captured from a stationary camera, they estimate the homography matrix to transform the detection bottom boxes into 2D BEV coordinates. By contrast, our method can be applied in moving camera requiring point annotations only. Nishimura et al. (2022) and Nishimura et al. (2023b) address the task of view birdification, which aims to convert hand-held or selfie camera views into BEV views. These methods explicitly assume that the observer's camera is parallel to the ground plane, meaning that the camera pose is known and fixed during inference. In contrast, our work tackles a more general and challenging problem, i.e., transforming from arbitrary camera viewpoints to BEV without requiring known camera poses during inference. Furthermore, unlike previous approaches, our Transformer based paradigm requires only foot location as supervisory signals and treats this problem as a coordinate regression problem.

## 3 OUR PROPOSED METHOD

In this section, we present the structure of the proposed multi-human image-to-BEV location framework, BEVCrowdLocator. In the following, we will elaborate the network design of collaborative learning in the image and BEV space, as well as the integration of proximity-aware suppression and standing point regression.

### 3.1 MULTI-HUMAN STANDING POINT REGRESSION

To yield accurate multi-human BEV positions, we approach the problem by predicting the positions where they are standing within the image, specifically the center point between their two feet, and then transform these positions into the BEV space via homography using the predicted camera pose.

**Utilizing Estimated Feet or Head Localization for BEV Projection.** While head detection offers higher visibility, individual height variations introduce significant errors in BEV projection via homography, which relies on accurate height estimation. In contrast, feet positioning circumvents this by adopting a fixed ground plane assumption, enabling height-agnostic projection. Though feet are often occluded, we leverage body part priors and neighbor context to resolve ambiguities. Experi-

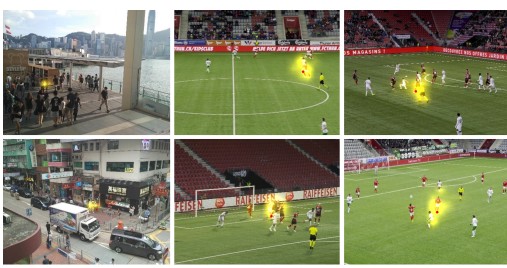 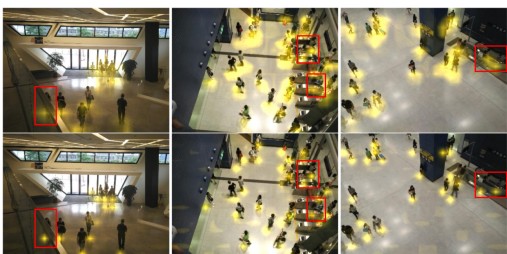

Figure 2: We visualize the cross-attention weight maps between a standing location query and the encoding embedding, indicating the salient proximal contexts that facilitate feet prediction (e.g., human body parts or neighboring persons).

Figure 3: Visualization on cross attention of the individual query and scene features. As observed, the focus on precise standing locations can be achieved via BEV contexts (the bottom row) comparing to that without BEV contexts (the top row).

mental results (Table 4 and Fig. 5 ) confirm that feet-based regression achieves better performance than head detection under occlusion.

**Network Structure.** We now detail our network architecture for multi-human standing point regression. As illustrated in Fig. 1, the primary network branch, Image View (IV) branch, comprises of a CNN-based feature extractor, a transformer encoder and decoder, and a point-based identity matcher. In contrast to BEV-Net (Dai et al., 2021) based on full CNNs, such a hybrid architecture, not only achieving a balance between model performance and efficiency, but also incorporating the proximal contexts (e.g., human body prior or neighboring interactions) through standing point queries which implicitly model their correlation. Specifically, given an input image $I$ ($I \in \mathbb{R}^{H \times W \times C}$), we first extracts the visual feature of multi-human scenes from a pretrained CNN backbone $\mathbf{F}$ (i.e., ResNet-50 (He et al., 2016)), which is then compressed and flattened into a one-dimensional sequence with positional embeddings (PE) before passing to the transformer encoder $\mathbf{T}_E$. $\mathbf{T}_E$ is a stack of encoding blocks, where each block consists of a self-attention layer and a feed-forward layer. Hence, the process can be expressed as $\mathbf{X}'_{IV} = \mathbf{T}_E(\text{Flatten}(\mathbf{F}(I)), \text{PE})$.

Subsequently, the transformer decoder $\mathbf{T}_D$ takes $\mathbf{x}_{IV}$ and the trainable embeddings of standing point queries $\mathbf{q}$ as input and, through cross-attention, generates decoded embeddings. $\mathbf{T}_D$ is a stack of decoding blocks. Each decoding block is mainly composed of three layers: (1) a query self-attention layer $F_{SA}$ that exerts interactions between the estimated standing point embeddings, each of which corresponding to a predicted point, enabling our model to handle duplicated prediction for proximal standing points; (2) a cross-attention layer $F_{CA}$ that correlates the standing point embeddings with the encoded features, modeling the relationship between the surrounding and humans, and (3) a feed-forward layer $F_{FF}$ that processes the outcomes of the preceding cross-attention layer to generate the queries $\mathbf{q}'$ for the next decoding block. The procedure can be formally described as: $\mathbf{q}' = F_{FF}(F_{CA}(F_{SA}(\mathbf{q}), X''_{IV}))$. As shown in Fig. 2, the standing location queries effectively infer the approximate foot locations according to the proximal contexts regarding human body and neighboring interactions, especially under conditions of compromised visibility.

Finally, the decoded embeddings are fed to standing point regression heads and classification heads, implemented as feed-forward networks (FFNs), for predicting foot position coordinates and their corresponding confidence scores. The predicted coordinates and confidence scores are further delivered to point-based identity matcher for supervision (please refer to Sec. 3.3).

## 3.2 COLLABORATIVE DUAL-SPACE LEARNING

Accurately predicting the targets' standing points that lie in the middle of two feet is not an easy task, as they are often affected by human poses or even occluded, resulting in noisy ground-truth annotations and uncertain prediction. Unlike IV features, although the supervisory signals of BEV coordinates may contain noise, they are not affected by visual occlusions and are less sensitive to individuals standing in close proximity to one another. To this end, we present an effective collaborative dual-space learning scheme, exploiting BEV contexts to assist standing location prediction.

To learn the features from the perspective of the BEV space, we establish an auxiliary network branch, i.e., BEV branch. As shown in Fig. 1, we employ a pretrained camera pose regression network (e.g. Dai et al. (2021)) that is capable of coarsely estimating the camera view pose from static input images $I$, denoted as $\Theta$. The estimated parameters of the camera view pose can be used to project the visual features of the input image $I$ (i.e., $X_{IV} = \mathrm{F}(I)$) to the corresponding BEV space via homographic transform, i.e., $X_{BEV} = \mathrm{Proj}(X_{IV}; \Theta)$. The projected features are then fed into a transformer encoder and decoder whose structure are identical to that of IV branch. Specifically, the BEV branch also embeds standing location queries and it is supervised by the annotated BEV coordinates. Since the IV features of targets are often incomplete due to compromised visibility, the encoded BEV features can be used to complement the IV features through BEV-to-image projection using a learnable MLP. Concurrently, the IV branch feeds back into the BEV branch, refining its encoding process and optimizing the camera pose estimation in the end-to-end manner. Overall, the proposed collaborative dual-space learning scheme facilitates mutual compensation between the IV and BEV features.

As shown in Fig. 3, this scheme integrates complementary contextual information from the BEV space into the IV space, enabling more precise prediction of standing coordinates. Notably, the highlighted regions exhibit the utilization of BEV contexts that infer the possible feet coordinates under severe occlusion.

### 3.3 PROXIMITY-AWARE POINT MATCHING

At the end of each network branch, we incorporate the point-based identity matcher to facilitate the standing point regression. Specifically, the head of each network branch infers $N$ predictions (i.e., the coordinates of the standing locations and the corresponding confidence scores), where $N$ is set to be significantly greater than the typical number of individuals in an image. On the training phase, in order to measure the gap between predictions and ground truth, we establish an effective correspondence between the predictions and ground truth. Therefore, we apply Hungarian matching algorithm (Kuhn, 1955) to realize the optimal bipartite graph matching between point predictions and ground truth. Hence, the cost of Hungarian matching, consisting of the location cost $\mathcal{C}_{sim}$ (i.e., L1 distance) and classification cost $\mathcal{C}_{cls}$ (i.e., focal loss (Lin, 2017)), while treating the unmatched predictions as background to bolster the model training. For any predicted point $\hat{y}_i$ with the confidence score $\hat{p}_i$ and the ground-truth point $y_j$, their matching cost is defined as:

$$\mathcal{C}_{match}(y_i, \hat{y}_j) = \mathcal{C}_{cls}(\hat{p}_j) + \omega \mathcal{C}_{loc}(y_i, \hat{y}_j) = -\alpha(1 - \hat{p}_j)^\gamma \log{(\hat{p}_j)} + \omega||y_i - \hat{y}_j||_1, \qquad (1)$$

where $\omega$ represents the weighting factor, which are set to 2.5. During training, the point matching loss $\mathcal{L}_{PM}$ is computed in the same way as the matching cost $\mathcal{C}_{match}$:

$$\mathcal{L}_{PM} = \mathcal{L}_{cls} + \omega \mathcal{L}_1 = -\alpha(1 - \hat{p}_{\sigma(i)})^\gamma \log{(\hat{p}_{\sigma(i)})} + \omega||y_i - \hat{y}_{\sigma(i)}||_1, \qquad (2)$$

where $\sigma(i)$ refers to the matched point for the $i$-th one.

Nevertheless, in multi-human scenes, the IV branch may produce incorrect predictions due to the complexity of regressing foot points. For instance, it may predict the two feet of the same individual as belonging to different people, or misidentify the feet of different individuals as those of the same person, resulting in duplicate predictions or missed predictions. To address this, we introduce a proximity-aware suppression operation before point-based identity matching. Specifically, we rank the pairwise distance of the predicted points with regarding their features/locations and then merge those close points. For the top K closest point pairs, we merge them into a single point, i.e., their center point, and update its confidence scores being their average scores, resulting in a new prediction set. With the new prediction set, we measure the loss following Eqs. (1) and (2).

On the other hand, since the estimated camera pose may be incorrect, we apply L2 loss to finetune the predicted camera height $\hat{h}$ and pitch angle $\hat{\theta}$ on the training phase:

$$\mathcal{L}_{\mathrm{pose}} = \lambda_{\mathrm{angle}}(\hat{\theta} - \theta)^2 + \lambda_{\mathrm{height}}(\hat{h} - h)^2. \qquad (3)$$

where the weighting factors $\lambda_{\mathrm{angle}}$ and $\lambda_{\mathrm{height}}$ are set to 2.0 and 0.02, respectively. $\theta$ and $h$ are the ground-truths.

In overall, as shown in Fig. 1, our proposed BEVCrowdLocator is trained subject to multiple objective functions, including the point matching losses on the image and BEV spaces ($\mathcal{L}_{PM}^{\mathrm{BEV}}$ and

Table 1: We evaluate the model performance for the task of social distancing compliance assessment on CityUHK-X-BEV.

| Method | Localization CD × 1 m ↓ | Local Risk IoU% ↑ | Global Risk MSE × $10^{-4}$ ↓ | Param. (M) | GFLOPs | Inf. Spd. (s) |
|---|---|---|---|---|---|---|
| Mask R-CNN (He et al., 2017) | 2.26 | 46.63 | 43.35 | 61.30 | 114.834 | 0.028 |
| CSRNet (Li et al., 2018) | 5.49 | 26.80 | 57.72 | 33.30 | 141.441 | 0.059 |
| DSSINet (Liu et al., 2019a) | 3.95 | 29.71 | 51.01 | 25.90 | 387.735 | 0.117 |
| CSP (Liu et al., 2019b) | 4.57 | 28.75 | 62.26 | 57.00 | 122.529 | 0.103 |
| BEV-Net (Dai et al., 2021) | 1.25 | 71.25 | 6.24 | 53.15 | 283.193 | 0.047 |
| CLTR (Liang et al., 2022a) | 1.70 | 53.15 | 8.94 | 43.45 | 135.847 | 0.017 |
| PET (Liu et al., 2023) | 1.64 | 54.02 | **1.11** | 51.65 | 859.680 | 0.050 |
| DINO (Zhang et al., 2022) | 1.30 | 59.80 | 10.03 | 64.70 | 124.204 | 0.024 |
| AlignDETR (Cai et al., 2024) | 1.36 | 63.65 | 2.45 | 64.70 | 124.204 | 0.024 |
| Ours | **0.57** | **83.64** | 2.05 | 80.77 | 151.975 | 0.021 |

$\mathcal{L}_{PM}^{\mathrm{IV}}$), the point matching loss enhanced by proximity-aware suppression $\mathcal{L}_{PM}^{\mathrm{IV+}}$ (see Sec. 4.4 for more analysis), as well as the camera pose estimation loss $\mathcal{L}_{\mathrm{Pose}}$, which can be expressed as:

$$\mathcal{L} = \mathcal{L}_{PM}^{\mathrm{BEV}} + \mathcal{L}_{PM}^{\mathrm{IV}} + \mathcal{L}_{PM}^{\mathrm{IV+}} + \mathcal{L}_{\mathrm{pose}}. \tag{4}$$

## 4 EXPERIMENTS

### 4.1 IMPLEMENTATION DETAILS

**Training configuration.** We implement our model using PyTorch in a workstation equipped with a NVIDIA RTX 3090 GPU. We employ ResNet-50 as the feature extractor, and both transformer encoder and decoder consist of six layers. The number of standing point queries is set to 500, and the batch size is configured to 8. The Adam (Kingma, 2014) optimizer is employed with a learning rate of 1e-4 to train the model. Additionally, a simple confidence threshold of 0.35 is applied to filter out the background class in the matcher. The value of $K$ in Top-K of the proximity-aware suppression is set as 2 to better strike a balance between model performance and computational overhead, since too large $K$ (e.g., $K >= 4$) tends to introduce incorrect merging and inaccurate shifts in predicted points, especially in dense scenarios.

**Cross-frame point association.** For associating the BEV locations across frames, we employ a trivial cross-frame association method, SORT (Bewley et al., 2016), to link the position of the same object across frames for all the comparison methods.

### 4.2 EVALUATION METRICS

For the task of SDCA, we employ the metrics of Chamfer distance, local risk error, and global risk error (Dai et al., 2021) for evaluation. 1) **Chamfer Distance (CD).** We estimate the localization error using the Chamfer distance, which measures the distance between the predicted location and the true location in real-world scale units (e.g. meters). 2) **Local Risk Error.** Local risk levels are represented as a heatmap, where higher values indicate a greater risk of infection, estimated by applying a Gaussian kernel on the location point map from the BEV space. Assuming that the infection risk is defined as simply the number of people within the safe distance to a person, the prediction quality is evaluated by intersection-over-union (IoU) w.r.t. ground-truth mask. 3) **Global Risk Error.** Global Risk Levels are defined as the total number of people that fail to maintain a minimum distance from each other in unit space. Global risk error is measured by MSE between estimated and ground-truth risks.

### 4.3 COMPARISON WITH STATE-OF-THE-ARTS

**Benchmark.** We evaluate the crowd BEV localization on CityUHK-X-BEV (Dai et al., 2021), which contains *static images* from 55 crowd scenes captured under various views of surveillance cameras. The training set consists of 2,503 images and the testset consists of 688 images. The

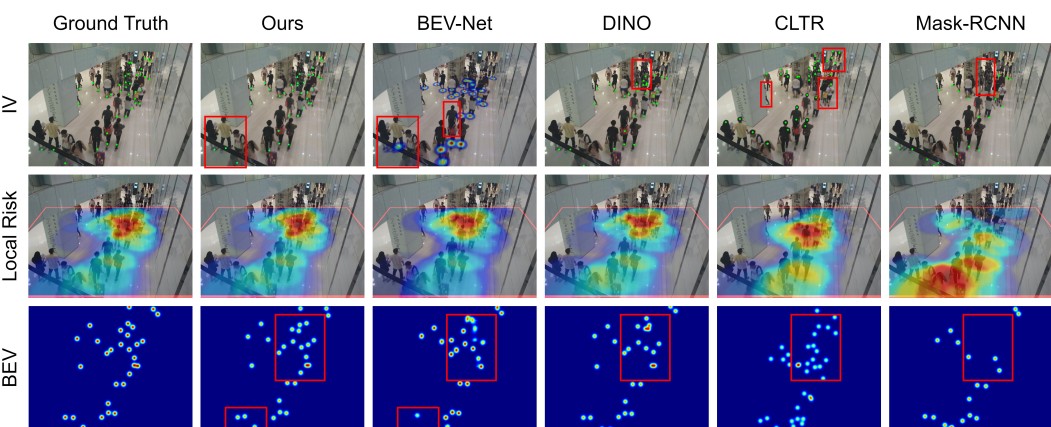

Figure 4: We visualize the multi-human BEV localization results on the CityUHK-X-BEV dataset. On the top and bottom rows, we show the localization results from the image and birds' eye views. On the middle row, we visualize the local risk map projected from BEV to IV, evaluating social distancing compliance.

Table 2: Evaluation for occluded targets on CityUHK-X-BEV.

|  | BEV-Net | CLTR | DINO | AlignDETR | Ours |
|---|---|---|---|---|---|
| Localiz. CD | 1.86 | 1.80 | 1.85 | 1.93 | **1.45** |

extrinsic parameters are known. We utilize this dataset to evaluate the ability of multi-human BEV localization for the task of social distancing compliance assessment (SDCA). **Camera calibration.** For this dataset, the homography matrix relies solely on the camera height $h$ and the camera pitch angle $\theta$. To predict these two variables, following Dai et al. (2021), we employ a camera pose prediction network, consisting of a feature extraction network based on VGG-16 (Simonyan & Zisserman, 2014) and three fully connected layers.

**Comparison Methods.** For comparison, in addition to the state-of-the-art BEV location method, BEV-Net (Dai et al., 2021), we follow their experiment protocol and compare our model against two types of baselines, detection-based methods and counting-based methods. For detection-based methods, we compared classic human detectors, including Mask R-CNN (He et al., 2017), CSPNet (Liu et al., 2019b), and DINO (Zhang et al., 2022). Like our approach, we integrate the pre-trained camera pose estimation network with these detectors. The bottom center of each bounding box was used as the standing locations. For counting-based methods, we compare our model with representative crowd counting networks including CSRNet (Li et al., 2018), DSSINet (Liu et al., 2019a), and CLTR (Liang et al., 2022a). In practice, we generate head distribution, and project them into the BEV space using the same camera pose estimation network. The vertical displacement between the head position and the ground plane is compensated by subtracting the average pedestrian height of 1.75m used in Kang et al. (2017) from the predicted camera height.

**Results.** Table 1 summarizes the performance of all methods on SDCA based on the evaluation metrics outlined in Sec. 4.2. Our proposed method significantly outperforms the other methods, achieving over a 50% reduction in Chamfer Distance and three times lower global risk error compared to the state-of-the-art BEV-Net (Dai et al., 2021), despite only using foot point annotations. The IoU of the local risk map estimated by our approach also shows advantage by a large margin for at least 12% comparing to the competing methods. In contrast to traditional methods, crowd counting models generally perform poorly, as they merely identify coarse head locations without the guidance of BEV contexts. Their simple strategy of subtracting a fixed default height (i.e., 1.75m (Kang et al., 2017)) for height estimation renders these approaches almost unreliable. Although the detection methods perform relatively better, they suffer from low recall in crowded scenes, resulting in numerous missed detections. In Fig. 4, we visualize the multi-human BEV localization results. As observed, for dense crowd where occlusion occurs, our model provides more precise location estimation in IV and BEV.

Table 3: Ablation on different components.

| CDSL | $\mathcal{L}_{PM}^{IV+}$ | Localization CD ↓ | Local Risk IoU% ↑ | Global Risk MSE ↓ |
|---|---|---|---|---|
| | | 0.62 | 79.31 | 2.56 |
| | ✓ | 0.62 | 80.29 | 2.48 |
| ✓ | | 0.58 | 82.30 | 2.25 |
| ✓ | ✓ | **0.57** | **83.64** | **2.05** |

Table 4: Evaluation of the training objective.

| Train. Obj. | Localization CD ↓ | Local Risk IoU% ↑ | Global Risk MSE ↓ |
|---|---|---|---|
| Head@IV | 1.49 | 54.34 | 6.13 |
| Feet@BEV | 1.17 | 82.51 | **1.65** |
| Feet@IV | **0.57** | **83.64** | 2.05 |

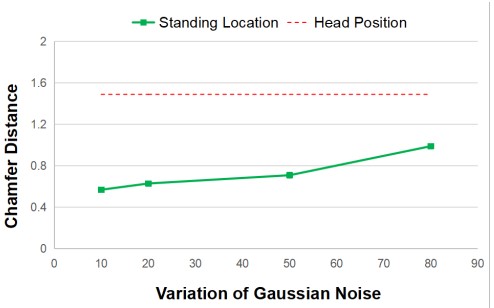

Figure 5: Model performance with feet location annotation contaminated by increasing scale of noises, as implies by the green line. For reference, the red dash line indicate the result with head annotation.

**Occlusion-Robust Evaluation.** To quantify performance under occlusion, we compute Chamfer Distance for occluded instances on CityUHK-X-BEV, leveraging their provided manual annotations of obscured individuals. As shown in Table 2, we evaluate the localization results under occlusion and our method achieves at least 19.4% lower CD than prior approaches, validating its superiority for these challenging cases in crowded scenarios.

**Model Efficiency.** We report the Floating Point Operations (FLOPs), inference speed, and number of parameters for our model with the input image size $384 \times 512$, as shown in Table 1. All tests were performed on a single NVIDIA RTX 3090 GPU. As observed, among the comparison methods, our model achieves the top-tier inference speed and efficiency even comparing to CNN-based models.

### 4.4 ABLATION STUDY

**Effectiveness of Proposed Components.** First of all, we validate the effectiveness of the components of our model including collaborative dual-space learning (CDSL) scheme and proximity-aware suppression. The ablation study results are reported in Table 3. Specifically, we evaluate the model performance without CDSL, which removes the BEV branch and preserve the primary branch only. As observed, with or without $\mathcal{L}_{PM}^{IV+}$, CDSL leads to obvious improvements, indicating that the BEV branch can extract context information from the BEV space that is not accessible in the IV space. On the other hand, proximity-aware suppression strengthens the point matching loss, enhancing the identification of closely spaced points. Thus, it does not greatly affect Chamfer Distance, but it results in significant discrepancies in risk predictions. It reflects that our approach is effective in addressing missed detections and duplicate predictions caused by occlusions.

**Feet versus Head Localization.** To evaluate this issue, we compare the metrics of three practices: 1) predicting the foot coordinates in the IV space; 2) directly regressing the BEV coordinates, and 3) predicting the head coordinates in the IV space and mapping them to BEV space with a fixed elevation of 1.75m. In Table 4, detecting heads in IV leads to suboptimal results, since it can hardly estimate individual heights precisely, resulting in severe coordinate deviation (see supplementary material for visualization results).

**Model Robustness Against Noisy Annotation of Standing Location.** However, during training phase, the major challenge lies in the uncertainty of manual annotation. To quantify the robustness of our model against the noisy annotation, as shown in Fig. 5, we train BEVCrowdLocator by adding Gaussian noises to standing location annotations (green line), while we use the results yielded by training with head annotations (red dashed line) for reference. It is evident that, with increasing noise scale, our model maintains stable localization performance and significantly better than the one supervised by head annotations, demonstrating the robustness of our model against noisy annotation.

**Analysis on Proximity-Aware Suppression.** To delve into the proximity-aware suppression on point matching loss, we add the suppression operation to the matching losses of the IV branch and BEV branch. As reported in Table 5, $\mathcal{L}_{PM}^{IV+}$ and $\mathcal{L}_{PM}^{BEV+}$ represent the point matching supervision

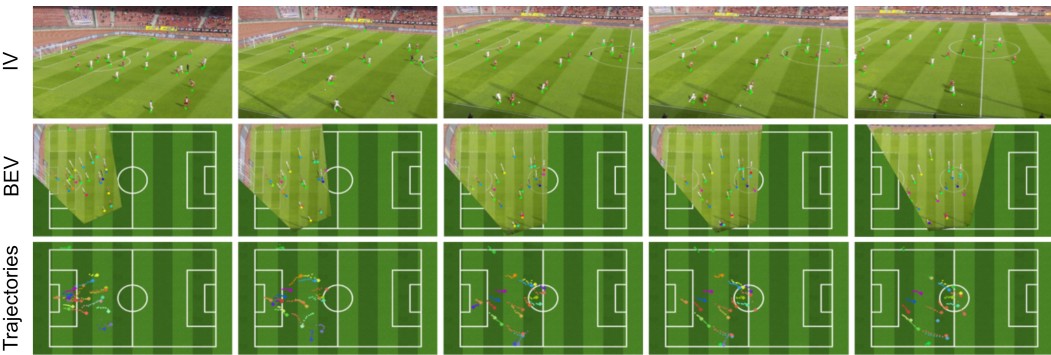

Figure 6: We showcase the results of applying our technique on soccer video clips. We demonstrate the players' localization results in the image view (IV) on the first row and BEV as well on the second row as the corresponding trajectories on the bottom row.

Table 5: Evaluation of the effectiveness of the proximity-aware suppression. CD, LR, and GR refer to Chamfer Distance, Local Risk Error, and Global Risk Error, respectively.

| $\mathcal{L}_{PM}^{IV}$ | $\mathcal{L}_{PM}^{IV+}$ | $\mathcal{L}_{PM}^{BEV+}$ | CD | LR | GR |
|---|---|---|---|---|---|
| ✓ | | | 0.58 | 82.30 | 2.25 |
| | ✓ | | 0.63 | 80.42 | 2.25 |
| ✓ | | ✓ | 0.60 | 81.27 | **1.92** |
| ✓ | ✓ | | **0.57** | **83.64** | 2.05 |

Table 6: Evaluation of the BEV tracking performance on SoccerNet without relying on pretrained detector. IV, BEV, and G.A. represent MOTA in the image view, birds' eye view, and goal area (with dense occlusion), respectively.

| Method | IV | BEV | G.A. |
|---|---|---|---|
| BEV-Net | 76.70 | 71.19 | 47.7 |
| CLTR | 59.13 | 69.41 | 46.3 |
| Ours | **85.24** | **80.60** | **51.6** |

enhanced by suppression, respectively. For reference, we also evaluate the supervision using the point matching loss on IV without suppression, i.e., $\mathcal{L}_{PM}^{IV}$. Comparing the results of the 1st, 2nd, and 4th rows, employing $\mathcal{L}_{PM}^{IV}$ and $\mathcal{L}_{PM}^{IV+}$ jointly achieves the best results, since $\mathcal{L}_{PM}^{IV}$ regularizes the false suppression. Moreover, comparing the results of the last two rows, enhancing the matching loss of the BEV branch decreases the location accuracy and local risk, yet leading to better global risk. It means that $\mathcal{L}_{PM}^{IV+}$ provides better prediction on individual positions, but $\mathcal{L}_{PM}^{BEV+}$ reduces the false prediction on BEV contexts, leading to better compensation to the IV features.

## 4.5 APPLICATION ON SPORTS VIDEO

We showcase the application of proposed method on sports video to demonstrate the potential of BEVCrowdLocator on the application of sports analysis. We employ SoccerNet (Cioppa et al., 2022) for evaluation. It consists of 200 30-second soccer match video clips (30 frames per second), which are split into training, validation, and test sets, as well as a separate challenge set. This dataset represents a highly dynamic broadcast scenario, characterized by rapid camera motion, significant viewpoint changes, and fast-moving players with diverse body poses. For quantitative evaluation, we compare it against BEV-Net (Dai et al., 2021) and CLTR (Liang et al., 2022a). To ensure a fair comparison, all the competing methods are restricted to using only point annotations of soccer players derived from both IV and BEV for training purposes, and they are not allowed to use pretrained human detectors. During test, these methods are required to predict the positions of players in the BEV space based on soccer videos. We employ the multi-object tracking metric MOTA (Bernardin & Stiefelhagen, 2008) as the evaluation metric. Additionally, all comparison methods employ identical cross-frame association method (i.e., SORT (Bewley et al., 2016)) to correlate the estimated BEV locations of players within video.

As shown in Table 6, our model's performance on the BEV tracking task in the SoccerNet benchmark surpasses other methods. Compared to the density map based method, BEV-Net, our model shows a significant lead in the term of MOTA. For the SOTA crowd localization method, CLTR, trained on head annotation, also shows suboptimal performance. As implies by MOTA in different views (i.e.,

IV and BEV), our method outperforms the competing methods. This advantage is attributed to the excellent localization capability of our point regression framework and the collaborative dual-space learning. Notably, our method achieves obvious advantage in the challenging regions near the goal (i.e., goal area) under dense occlusions, since our dual-space learning leverages the BEV features to compensate for occlusions occurred in the IV space. Additionally, as observed in Fig. 6, we showcase the BEV tracking results using our approach from the perspectives of image view and birds' eye view as well as the trajectories. We also provide the results in the supplemented video, which visually demonstrates the performance of our model in such high-motion scenes.

## 5 CONCLUSION

This paper addresses the challenges of multi-human BEV localization in surveillance and sports analytics, focusing on occluded visibility and the need for proximal context. Our key contribution is a collaborative dual-space learning scheme that integrates BEV context with image-space inference, enhancing model robustness against occlusions. By introducing standing point queries as trainable embeddings, our model learns proximal context and distinguishes between closely situated individuals' foot positions. We also incorporate proximity-aware suppression into standing point regression, further enhancing model performance under severe occlusion. Validated on CityUHK-X-BEV for social distance measurement and SoccerNet for sports analysis, our approach demonstrates significant potential in advancing multi-human BEV localization tasks. However, intense camera motion and zooming may introduce pose estimation errors, which can in turn affect localization accuracy. As future work, integrating an end-to-end and robust camera calibration module into the framework could further enhance overall performance. In addition, efforts towards constructing larger-scale datasets with more precise annotations and greater scene diversity represent a promising direction, as such datasets would provide richer and more challenging scenarios for training and evaluating advanced models.

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
