# OpenReview forum: "Pinpointing Crowd in Bird's Eye View via Proximal Contexts"
_ICLR.cc/2026/Conference — Submitted to ICLR 2026_

### Official Review · Reviewer_uzQw · 2025-10-29

**Soundness:** 4
**Presentation:** 4
**Contribution:** 3
**Rating:** 8
**Confidence:** 4

**Summary:**

1. This paper focuses on the challenging problem of multi-human localization in Bird's-Eye-View(BEV) from monocular images, especially in the case of crowded and occluded scenarios like sports fields or surveillance scenes.
2. The authors propose BEVCrowd-Locator, a novel framework that formulates the task as a direct regression of individual standing points in image space, and converts to BEV space based on the predicted camera poses.
3. The proposed dual-space learning scheme and the proximity-aware suppression mechanism are useful for mitigating duplicate predictions of the same individual.
4. The result on the CityUHK-X-BEV dataset shows the effectiveness of the proposed framework.

**Strengths:**

1. The proposed framework is novel. The dual-space learning scheme with CNN-Transformer hybrid network architecture, as well as the proximity-aware suppression mechanism, contributes to the improvement on the benchmark.
2. The ablation studies are clear and well-designed. The ablation studies provide clear evidence for the necessity of each proposed component.

**Weaknesses:**

1. While the state-of-the-art method for crowd analysis is BEV-Net, it was proposed in 2021, thus it would be nice to compare the proposed method with more recent and general proposed detectors like PairDETR.
2. The image-to-BEV projection relies on the given camera poses estimated by the network. The framework may be limited when cameras are moving or changing. It would be nice if authors could discuss the related limitations.

**Questions:**

Please refer to the Weaknesses section.

---

> ### Author Response · Authors · 2025-11-27
> **Response to Reviewer uzQw (1/2)**
>
> We sincerely thank the reviewer for the positive and encouraging evaluation of our work. We truly appreciate the reviewer's recognition of the novelty of our BEVCrowd-Locator framework, the effectiveness of the dual-space learning scheme and proximity-aware suppression mechanism, and the clarity and rigor of our ablation studies. We are also grateful for the constructive suggestions, which help us further improve the completeness and clarity of our paper.
>
> **Weakness 1: The comparative methods are relatively outdated.**
>
> We fully acknowledge this concern and thank the reviewer for pointing it out. Our selection of baselines was primarily guided by task relevance, as few recent works directly address multi-human BEV localization.  We would like to clarify that BEV-Net is the most relevant approach to our task and currently represents the SOTA method in this domain. Since object detection methods and crowd counting methods collaborate with naive view transformation may also achieve similar purpose as our task, we involved several representative object detection methods for comparison.
>
> We also appreciate the reviewer's suggestion to include comparisons with more recent detectors such as PairDETR (CVPR'24) [1]. PairDETR is indeed an advanced method that jointly performs face–body detection and association. However, our main evaluation datasets—CityUHK-X-BEV and SoccerNet—do not contain face annotations, making it difficult to directly adapt PairDETR to our setting. Considering that PairDETR employs DETR as its underlying detector, we instead conducted comparisons with the latest DETR variant, AlignDETR. Our method still outperforms AlignDETR by a clear margin, further validating the superiority and robustness of our approach.
>
> | Method |  Avenue | Localization CD  |  Local Risk IoU | Global Risk  MSE  |
> | ------------ | ------------ | ------------ | ------------ | ------------ |
> | CLTR[2] | ECCV'22 |1.70  | 53.15  | 8.94 |
> | PET[3] | ICCV'23  |  1.64 |  54.02 |  1.11 |
> | DINO[4] | ICLR'23 | 1.30 | 59.80 | 10.03 |
> |  AlignDETR[5] |  BMVC'24 | 1.36  |  63.65 |  2.45 |
> | Ours |  - |  0.57 | 83.64  |  2.05 |

---

> ### Author Response · Authors · 2025-11-27
> **Response to Reviewer uzQw (2/2)**
>
> **Weakness 2: Discuss the limitations of image-to-BEV projection with dynamic cameras.**
>
> We agree that the performance of the BEV transformation process may be influenced by camera motion. For instance, when the camera zooms in a small local region, the limited visible scene provides insufficient contextual cues for accurate pose estimation, which may lead to larger pose errors and consequently slight localization deviations. We will elaborate on this limitation more thoroughly in the revised paper and discuss potential solutions, such as integrating video-based pose stabilization techniques to enhance robustness under camera movement.

---

> ### Author Response · Authors · 2025-11-27
> **Official Comment by Authors**
>
> # References
> [1] Ali, Ammar, et al. "PairDETR: Joint detection and association of human bodies and faces." Proceedings of the IEEE/CVF Conference on Computer Vision and Pattern Recognition. 2024.
>
> [2] Liang, Dingkang, Wei Xu, and Xiang Bai. "An end-to-end transformer model for crowd localization." European Conference on Computer Vision. Cham: Springer Nature Switzerland, 2022.
>
> [3] Liu, Chengxin, et al. "Point-query quadtree for crowd counting, localization, and more." Proceedings of the IEEE/CVF international conference on computer vision. 2023.
>
> [4] Zhang, Hao, et al. "Dino: Detr with improved denoising anchor boxes for end-to-end object detection." arXiv preprint arXiv:2203.03605 (2022).
>
> [5] Cai, Zhi, et al. "Align-DETR: Enhancing end-to-end object detection with aligned loss." arXiv preprint arXiv:2304.07527 (2024).

---

### Official Review · Reviewer_LAAS · 2025-10-30

**Soundness:** 3
**Presentation:** 3
**Contribution:** 2
**Rating:** 4
**Confidence:** 4

**Summary:**

This paper proposes a framework termed BEVCrowdLocator, for multi-human BEV standing localization. Through dual-space collaborative learning between image space and BEV space, the model achieves precise prediction of individual standing points under monocular vision, thereby addressing severe occlusion issues in dense crowd scenarios. Key innovations include: Standing point regression: Incorporating trainable point queries within the Transformer architecture to model spatial relationships between adjacent individuals, enabling foot-level localisation based on neighbourhood context. Proximity-aware point matching: Merging duplicate detection points to enhance localisation robustness. Dual-space collaborative learning mechanism: During training, BEV features compensate for occlusion information in the image space, thereby improving overall prediction accuracy.

**Strengths:**

The approach exhibits a certain degree of innovation, as this paper proposes a novel architecture and corresponding design mechanism that together constitute a relatively well-structured overall framework. The experimental validation is relatively comprehensive, with the paper comparing multiple methods and conducting ablation experiments to demonstrate the effectiveness of the proposed approach. It also possesses a certain level of generalizability, as experiments were conducted on two distinct datasets, both yielding favorable results

**Weaknesses:**

1. Dual-space collaborative learning essentially extends the concept of image–BEV feature alignment (including feature projection and fusion), similar to existing approaches such as BEVFusion and the BEVFormer series. The proposed CNN–Transformer architecture follows a well-established framework and does not exhibit particular novelty. The authors are encouraged to provide a more in-depth analysis of the innovative aspects of their proposed framework, specifically clarifying its most significant distinctions or advancements compared with existing methods.

2. The comparative methods employed in the experiments are relatively outdated, with only one originating from 2024 and most being based on approaches from 2022 or earlier. As a result, the comparisons do not sufficiently demonstrate the superiority of the proposed method. In addition, the experimental dataset is relatively small in scale, and the effectiveness of the proposed approach should be further verified on large-scale, real-world scenarios. Moreover, the ablation studies lack a detailed analysis of how varying the range of Top-K values influences the model’s performance.

**Questions:**

1. Since both the overall concept of the collaborative framework and the CNN-Transformer architecture are established approaches lacking particular novelty, the authors are encouraged to focus on analysing the innovative aspects of the proposed framework. In particular, they should clarify the key distinctions and innovations that differentiate it from existing methods.

2. The comparative methods employed in the experiments are relatively outdated, and the authors are requested to clarify the rationale behind this selection. In addition, the experimental results should be supplemented with comparisons to more recent methods. The experimental dataset is also relatively small in scale, whereas the network model contains a large number of parameters, which raises concerns about potential overfitting. Therefore, experiments on larger-scale datasets are recommended to further validate the effectiveness of the proposed method. Moreover, although the paper introduces the concept of Top-K, it does not provide corresponding experiments analysing the impact of different value ranges within this setting.

---

> ### Author Response · Authors · 2025-11-27
> **Response to Reviewer LAAS (1/3)**
>
> We sincerely thank the reviewer for the constructive and thoughtful feedback. We appreciate the recognition of our work's **innovation**, **solid framework design**, and **comprehensive experiments**. Below, we provide detailed clarifications and responses to each of the reviewer's concerns.
>
> **Weakness 1: The dual-space collaborative learning mechanism appears conceptually similar to existing BEV feature alignment methods such as BEVFusion and BEVFormer. Please clarify the novelty and unique contributions.**
>
> We appreciate this insightful observation.  While our framework shares the core idea of aligning multi-view image features with BEVFusion and BEVFormer, it differs from and innovates upon prior work in several key aspects:
> - We introduce an interactive rather than unidirectional feature-space transformation scheme. Existing BEV fusion frameworks, such as BEVFusion and BEVFormer, primarily project multi-view images into the BEV space and then perform predictions based on BEV features. In contrast, our model establishes bidirectional collaboration between IV and BEV spaces through dual-space mappings (IV-to-BEV and BEV-to-IV), allowing BEV features to enhance perception in the image-view space.
> - The proposed paradigm is based on direct coordinate regression instead of density-map reliance. Unlike prior representative methods such as BEV-Net, which depend on traditional density map based approaches - intrinsically limiting their ability to localize individuals accurately in dense crowds - our method leverages the strong representation power of Transformer to directly regress precise individual coordinates.
> - We present proximity-aware suppression for dense groups of targets. We introduce a proximity-aware matching loss that effectively addresses repeated or overlapping detections. This mechanism is innovative and crucial for multi-person localization, whereas previous BEV methods rarely handle redundancy among similar objects.
>
> We will revise the manuscript to more explicitly highlight these unique contributions and clarify the distinctions between our approach and existing methods.

---

> ### Author Response · Authors · 2025-11-27
> **Response to Reviewer LAAS (2/3)**
>
> **Weakness 2: The comparative methods are relatively outdated, with limited inclusion of recent approaches, and the dataset scale is small.**
>
> We fully acknowledge this concern and thank the reviewer for pointing it out. Our selection of baselines was primarily guided by task relevance, as few recent works directly address multi-human BEV localization.  We would like to clarify that BEV-Net is the most relevant approach to our task and currently represents the SOTA method in this domain. On the other hand, object detection methods and crowd counting methods collaborate with naive view transformation may also achieve similar purpose as our task, so we involved several representative object detection methods for comparison.
>
> Regarding the dataset size, we would like to clarify that the SoccerNet dataset contains 200 clips of 30-second football match videos (30 frames per second), amounting to approximately 180,000 images. These images capture real-world scenarios with significant viewpoint variations and severe occlusions, fully satisfying the requirements for evaluation. We will also emphasize this point in the experimental section of the paper.

---

> ### Author Response · Authors · 2025-11-27
> **Response to Reviewer LAAS (3/3)**
>
> **Weakness 3: The ablation study lacks analysis on how different Top-K ranges affect model performance.**
>
> Thank you for pointing this out. We have conducted additional experiments with different values of Top-K, which show that the results are not significantly affected by this parameter. Specifically, on the CityUHK-X-BEV dataset, the Chamfer Distance is 0.57 for K = 2 and 0.59 for K = 3. We believe that a smaller K can better strike a balance between model performance and computational overhead, while too large K  (e.g., K>=4) tends to introduce incorrect merging and inaccurate shifts in predicted points, especially in dense scenarios. Therefore, we adopt K = 2 for both datasets. We will provide a more detailed explanation and analysis in the experimental section.

---

### Official Review · Reviewer_8r8w · 2025-10-31

**Soundness:** 3
**Presentation:** 3
**Contribution:** 3
**Rating:** 4
**Confidence:** 4

**Summary:**

This paper addresses the challenge of multi-human Bird's-Eye View localization from monocular images in crowded scenes with severe occlusions by proposing a novel method named BEVCrowd-Locator. Traditional approaches relying on coarse density maps or head detection—which is sensitive to height variations—struggle to achieve precise individual-level localization in dense crowds. To overcome this, the work reframes the task as a standing point (the center between the feet) regression problem and introduces a unified collaborative dual-space learning framework. This framework consists of a primary image-view branch and an auxiliary BEV branch, which compensate and provide feedback to each other via learnable mappings. By leveraging contextual information from the BEV space that is unaffected by visual occlusions, the model significantly enhances the robustness of image-space inference.
The core innovation lies in the introduction of trainable standing point queries. These embeddings explicitly learn proximal contexts, such as human body priors and spatial relationships between neighboring individuals, through the attention mechanisms in the Transformer architecture. This enables the model to effectively infer occluded foot locations and distinguish between adjacent persons. Furthermore, the paper proposes a proximity-aware suppression mechanism that aggregates redundant, overly close predicted points during regression, effectively mitigating duplicate detections of the same individual and further improving localization accuracy.
Comprehensive experiments validate the approach: On the CityUHK-X-BEV surveillance benchmark for social distancing compliance assessment, the method significantly outperforms state-of-the-art techniques, reducing the Chamfer Distance by over 50%. On the SoccerNet soccer video dataset, utilizing only point annotations and a simple cross-frame association module, the method achieves smooth BEV-based multi-human tracking, demonstrating exceptional performance especially in densely occluded areas like the goal box, highlighting its potential for dynamic sports analysis. Extensive ablation studies confirm the effectiveness of each proposed component. In summary, this work provides a powerful and efficient solution for precise multi-human BEV localization in complex scenes by cleverly leveraging collaborative information from both image and BEV spaces and modeling proximal relationships between individuals.

**Strengths:**

This paper presents the BEVCrowdLocator, a model aimed at multi-human localization in crowded environments, particularly addressing occlusion challenges. The core innovation is the use of dual-space collaborative learning, which combines image and Bird’s-Eye View (BEV) spaces to improve localization accuracy. The model introduces a context-driven feet localization method, utilizing proximal context via standing point queries to distinguish individual positions even under limited visibility. Additionally, a proximity-aware suppression mechanism is incorporated to reduce redundancy in predictions.
The paper is technically solid, with clear descriptions of the model architecture and its components. It employs transformers in both image and BEV spaces, demonstrating a strong understanding of the task at hand. Experimental results, including evaluations on CityUHK-X-BEV and SoccerNet, show that the proposed method outperforms existing approaches in terms of accuracy, particularly in occluded scenarios, while maintaining reasonable computational efficiency.
The structure of the paper is logical, with a clear explanation of the methodology and well-organized experimental sections. The approach is well-validated through extensive experimentation, and the model’s potential applications extend beyond surveillance and sports analytics, with relevance to areas such as autonomous driving and robotics. Overall, the paper contributes a meaningful advancement to multi-human localization in complex environments.

**Weaknesses:**

While the paper presents a strong contribution, there are a few areas where further improvements could be made. First, the discussion of the model's limitations is brief, particularly regarding its performance under extreme occlusions or in highly dynamic environments. A more detailed analysis of such scenarios would help clarify the model's robustness. Additionally, the evaluation is mainly conducted on two datasets (CityUHK-X-BEV and SoccerNet), and more diverse datasets representing varied crowd behaviors and environmental conditions would better assess the model’s generalization capabilities.
Another weakness is the occlusion handling in moving scenarios; while the paper addresses occlusion in static settings, it lacks discussion on performance with moving cameras or in dynamic environments. The model's scalability for handling large crowds also needs more exploration, especially regarding performance when the number of individuals increases.
Furthermore, while the paper compares feet localization with head detection in occluded settings, a deeper comparison of the two methods under varying levels of occlusion could provide a clearer understanding of the trade-offs. Finally, real-time performance and hyperparameter sensitivity are not fully addressed. Testing the model in real-time scenarios and evaluating the impact of different hyperparameters would provide a more complete picture of its practical deployment and tuning. Addressing these points would further enhance the paper’s impact and applicability.

**Questions:**

To further improve the paper, I have a few questions and suggestions for the authors. First, could you provide additional details on how the BEVCrowdLocator model performs under extreme occlusion scenarios, such as when individuals are densely packed or largely obscured by other objects? Secondly, while the paper evaluates the model in static settings, how does it handle moving cameras or dynamic viewpoints? A discussion of performance with changing perspectives would be valuable. Third, the paper demonstrates good results with smaller crowds, but how does the model scale when dealing with larger crowds in real-world scenarios? It would be helpful to explore scalability in dense crowds. Additionally, while feet localization is emphasized, could you provide a deeper comparison with head detection, especially in more complex scenarios with high occlusion? A more thorough analysis of this trade-off would be beneficial. Regarding real-time performance, could you share how the model performs on live video feeds or in resource-constrained environments, such as edge devices? Lastly, it would be useful to include an ablation study or analysis of hyperparameter sensitivity, particularly regarding the number of standing point queries and transformer settings. Addressing these aspects would improve the understanding of the model’s limitations, robustness, and potential real-world deployment.

---

> ### Author Response · Authors · 2025-11-27
> **Response to Reviewer 8r8w (1/5)**
>
> We sincerely thank the reviewer for the positive assessment of our work, recognizing the **technical soundness**, **clear presentation**, and **contributions of BEVCrowd-Locator**. We appreciate the thoughtful comments and constructive suggestions, which will help us further improve the paper. Below, we address each concern in detail.
>
> **Weakness 1: The paper could include more discussion or analysis of performance under extreme occlusions or in highly dynamic environments.**
>
> We sincerely appreciate this valuable suggestion. To further validate the effectiveness of our method under occluded scenarios, we conducted additional experiments on high-occlusion subsets of both datasets. On the CityUHK-X-BEV dataset, we evaluated a manually annotated occlusion subset, where our method achieved a 8.2% improvement in precision over BEV-Net (see Supplementary Table 1). On the SoccerNet dataset, within the goal area - a region frequently affected by dense occlusions - our approach reduced the Chamfer Distance by 22% compared to the next best method (see Table 5). It is also worth emphasizing that the SoccerNet dataset represents a highly dynamic broadcast scenario, characterized by rapid camera motion, significant viewpoint changes, and fast-moving players with diverse body poses. These factors naturally make it a challenging and dynamic setting. We refer the reviewer to the supplementary video results on SoccerNet, which visually demonstrate our model's performance in such high-motion scenes. We have highlighted this point explicitly in the revised paper.

---

> ### Author Response · Authors · 2025-11-27
> **Response to Reviewer 8r8w (2/5)**
>
> **Weakness 2: The scalability to large crowds and dense real-world scenarios could be explored more deeply.**
>
> Thank you for highlighting this important point. The standing point query mechanism is specifically designed to maintain scalability by using a fixed number of learnable queries that dynamically attend to individual points rather than predicting dense maps. Moreover, we set the number of queries to the maximum number of targets that may occur in the scenario to ensure stable performance. Due to the lack of datasets for evaluating BEV localization of large/dense crowd, we will consider it as the major future direction. We appreciate the reviewer's advice.

---

> ### Author Response · Authors · 2025-11-27
> **Response to Reviewer 8r8w (3/5)**
>
> **Weakness 3: Real-time performance and hyperparameter sensitivity are not fully addressed.**
> We sincerely thank the reviewer for this valuable comment. While real-time performance is not the primary focus of our work, we nonetheless report the inference speed on an RTX 3090 GPU in Table 1 for reference.
>
> Regarding hyperparameter sensitivity, we include relevant analyses in the supplementary material:
> - Supplementary Table 3: Analysis on the threshold of proximity-aware suppression;
> - Supplementary Table 5: Analysis on the weighting factor of the point matching loss.

---

> ### Author Response · Authors · 2025-11-27
> **Response to Reviewer 8r8w (4/5)**
>
> **Weakness 4: More diverse datasets representing varied crowd behaviors and environmental conditions would better assess the model's generalization capabilities.**
>
> We thank the reviewer for the insightful comment. Our current evaluation focuses on CityUHK-X-BEV and SoccerNet, where CityUHK-X-BEV represents static surveillance scenarios, while SoccerNet reflects highly dynamic sports environments. However, we acknowledge that more diverse datasets would further strengthen the validation of our approach. At present, there is a lack of suitable large-scale public datasets for multi-human BEV localization. In future work, we plan to explore or construct datasets with more diverse scenes, and potentially annotate new benchmarks to better evaluate and generalize our method across different environments.

---

> ### Author Response · Authors · 2025-11-27
> **Response to Reviewer 8r8w (5/5)**
>
> **Weakness 5: A deeper comparison of the two methods under varying levels of occlusion could provide a clearer understanding of the trade-offs.**
>
> We appreciate the reviewer's sharp insight and valuable suggestion. Quantitatively evaluating the effect of different levels of occlusion is indeed challenging, as it requires fine-grained annotation of occlusion degrees within the dataset, which is both complex and labor-intensive. In future work, we plan to construct or extend datasets with explicit occlusion-level annotations to enable more systematic analysis and comparison between different localization strategies under varying occlusion conditions.

---

### Official Review · Reviewer_oK74 · 2025-11-02

**Soundness:** 3
**Presentation:** 1
**Contribution:** 2
**Rating:** 4
**Confidence:** 3

**Summary:**

This paper focuses on multi-human standing (feet) localization in crowded scenes. Specifically, it introduces a unified dual-space collaborative learning with a dual-branch network. For the spaces, it uses both image/BEV spaces via image-to-BEV/BEV-to-BEV transformation with homography. For architecture, the primary branch (a hybrid CNN-transformer network) regresses standing locations in the image space. The auxiliary branch estimates standing locations in the BEV space and provides features for the primary branch. Lastly, proximity-aware suppression is used to distinguish close points to enhance point matching loss. Experiments on CityUHK-X-BEV and SoccerNet show that the proposed method outperforms existing methods.

**Strengths:**

- The usage of both image/BEV spaces via image-to-BEV/BEV-to-BEV transformation can effectively combine the advantages of different spaces. This is a high-efficiency framework.

**Weaknesses:**

1. The input is confusing.
- L.28-38 emphasizes that this paper focuses on BEV perception, which is vague. L.178 only states that the input is an image. Based on Figs.1-4, the input is image view (IV) maps and only the results are shown in BEV. The primary branch is the IV branch, which is also confusing. Because it is not clear about the usage of BEV-to-image for the input (Fig.1).
- Existing sotas are not designed for BEV, which may raise concerns about fairness. It would be better to compare with BEV methods (L.124-136) instead of method modification (L.320).

2. The difference compared to BEV-Net should be discussed in more depth.
- BEV-Net also uses the dual-space design with homography. The main difference seems to be the usage of Transformer (L.41).
- the paper discusses about feet vs head localization (L.54 and 398). However, the methods that use the head for detection are limited. BEV-Net uses head and feet for detection.

3. This paper highlights that the new formulation can handle occlusion in crowded scenes (L.14). However, there is no in-depth analysis. Quantitative or localization results like Fig.4 cannot reveal the reasons. It would be better to show the intermediate results like attention map in two spaces for occlusion.

**Questions:**

See Weaknesses.

---

> ### Author Response · Authors · 2025-11-27
> **Response to Reviewer oK74 (1/3)**
>
> We thank the reviewer for recognizing our **high-efficiency framework** and **SOTA performance**. We appreciate your support and constructive suggestions and address your concerns as follows.
>
> **Weakness 1: The input is confusing. The paper focuses on BEV perception, but the primary branch of the framework is the IV branch. The transformation from BEV to the IV branch is also unclear. Moreover, the method selection for the comparative experiments is unfair.**
>
> Thank you for your valuable suggestion. We will provide more explanation and clarification for the proposed framework pipeline and the selection of comparison methods:
> - _About the model input._ To clarify, our overall framework only takes images as input - no camera poses are required as input, since they are predicted by the network itself.
> - _About BEV perception._ Sorry for the misunderstanding. Our work aims to localize individual people for multi-human scenes in a bird's-eye view from perspective images. We have rephrased it to make it clear.
> - _About BEV-to-IV transformation._ The detailed transformation process is as followed. In the BEV branch pipeline, the predicted camera poses are first used to transform IV-space features into the BEV space (L.213). These features are then processed by the BEV encoder and subsequently mapped back into the IV space to enhance the perception capability of the IV branch (L.216). Finally, the BEV perception results are obtained by transforming the IV branch's prediction into the BEV view using the predicted camera poses (L.149).
> - _About comparison methods._ We would like to clarify that BEV-Net is the most relevant approach to our task and currently represents the SOTA method in this domain. Regarding the BEV crowd analysis methods mentioned in Lines 124–136, the works such as [1] and [2] address the "view birdification" task, which aims to convert hand-held or selfie camera views into BEV views. These approaches explicitly assume that the observer's camera is parallel to the ground plane, meaning that the camera pose is known and fixed during inference. In contrast, our task targets a more general and challenging problem - transforming from arbitrary camera viewpoints to BEV space without requiring known camera poses during inference. The codes of both methods were not released as well. We sincerely appreciate your insightful feedback and will provide a more detailed and comprehensive comparison and discussion with these related BEV methods in the Related Work section of the revised paper. On the other hand, object detection methods and crowd counting methods collaborate with naive view transformation may also achieve similar purpose, so we involved several representative object detection methods for comparison.

---

> ### Author Response · Authors · 2025-11-27
> **Response to Reviewer oK74 (2/3)**
>
> **Weakness 2: The difference from BEV-Net should be discussed in more depth.**
>
> We appreciate the advice. First of all, it is important to clarify that BEV-Net was originally designed for social distance risk estimation, where obtaining precise individual locations is not the primary goal. In contrast, our work aims to facilitate crowd behavior analysis, which requires highly accurate localization of each individual in the scene. Second, from the technical view, BEV-Net relies on density maps for supervision (same as crowd counting), which inherently limits its ability to capture precise individual positions in densely crowded scenes. On the contrary, we leverage the powerful representation capability of Transformers to directly regress the exact location coordinates of each individual.  Third, BEV-Net requires both head and feet annotation for supervision. We discovered that, in our paradigm, adopting feet location only as supervisory signals is enough since predicting individual heights is highly uncertain and error-prone, which would lead to inaccuracies in the derived BEV perception results and error accumulation during the transformation process. Last, unlike BEV-Net that simply projects IV-space features into the BEV space for prediction, our framework strengthens bidirectional interaction and collaboration between two spaces (IV-to-BEV and BEV-to-IV). Since head and foot locations overlap when projected to the bird's-eye view, our bidirectional design lessens the dependency on annotated head positions. We have clarified this in the revision.

---

> ### Author Response · Authors · 2025-11-27
> **Response to Reviewer oK74 (3/3)**
>
> **Weakness 3: There is a lack of in-depth analysis of the improvement of occluded scenes and the visualization of intermediate results.**
>
> We would like to thank the reviewer for the thoughtful and thorough comments on our paper. In Table 5 of the main text and Table 1 of the supplementary materials, we present quantitative experiments conducted on the goal area - where occlusion frequently occurs - and on the annotated occlusion subset of the CityUHK-X-BEV dataset, respectively. These results demonstrate that our method effectively improves the performance in occluded scenes to a certain extent.
>
> Additionally, in Figure 3 of the paper, we visualize the cross-attention maps between the standing point queries and the feature maps, with and without the incorporation of BEV-space interaction. The visualization shows that, thanks to the BEV representation where the head and feet are projected onto the same spatial point, the BEV-space information enables the standing point queries to focus more precisely on the true standing locations. This clearly indicates that the BEV-space features contribute to achieving more accurate predictions. We will provide a more detailed and specific discussion in the revised manuscript.

---

> ### Author Response · Authors · 2025-11-27
> **Official Comment by Authors**
>
> # References
>
> [1] Mai Nishimura, Shohei Nobuhara, and Ko Nishino. Viewbirdiformer: Learning to recover ground-plane crowd trajectories and ego-motion from a single ego-centric view. IEEE Robotics and Automation Letters, 8(1):368–375, 2022.
>
> [2] Mai Nishimura, Shohei Nobuhara, and Ko Nishino. View birdification in the crowd: Ground-plane localization from perceived movements. International Journal of Computer Vision, 131(8):2015–2031, 2023b

---

### Meta-Review · Area_Chair_RpcN · 2026-01-07

**Summary:**

The paper presents BEVCrowd-Locator, a framework for multi-human standing localization using monocular images. The core innovation is the use of dual-space collaborative learning, which combines image and Bird’s-Eye View (BEV) spaces to improve localization accuracy. By combining a CNN-Transformer hybrid network with trainable standing point queries and a proximity-aware suppression mechanism, the method aims to resolve occlusions and improve localization accuracy.

The reviewers generally agree that the dual-space approach is technically sound and that the experimental results on CityUHK-X-BEV and SoccerNet are promising. However, three out of four reviewers initially rated the paper as "marginally below the acceptance threshold". Their primary concerns involve the difference between the proposed method and BEV-Net, unclear implementation details, the use of outdated baselines (e.g., comparing mainly against 2021/2022 methods), and a lack of discussion on dynamic scenarios (moving cameras/extreme density).

**Reviewer Concerns:**

The reviewers’ primary concerns lie in whether the technical distinctions between this work and existing BEV literature are sufficiently significant. For instance, Reviewer LAAS explicitly notes that the "proposed CNN–Transformer architecture follows a well-established framework and does not exhibit particular novelty." While the authors provide detailed elaborations regarding implementation details and analysis in dynamic scenes, some other concerns remain unsolved.

**Reviewer Scores:**

Reviewer 8r8w mention evaluations on more datasets and more crowded scenes, but the authors do not provide further results or other alternatives, so the rating will likely not change. Reviewer LAAS mention the main technical differences and point out the lack of novelty. The rating will probably not change either.

---

### Decision · Program_Chairs · 2026-01-26

Reject